# MESHDIFFUSION:
# SCORE-BASED GENERATIVE 3D MESH MODELING

**Zhen Liu**[1,2,*]**, Yao Feng**[2,3]**, Michael J. Black**[2]**, Derek Nowrouzezahrai**[4]**, Liam Paull**[1]**, Weiyang Liu**[2,5]

[1]Mila, Université de Montréal   [2]Max Planck Institute for Intelligent Systems - Tübingen
[3]ETH Zürich   [4]McGill University   [5]University of Cambridge

Project Page: meshdiffusion.github.io

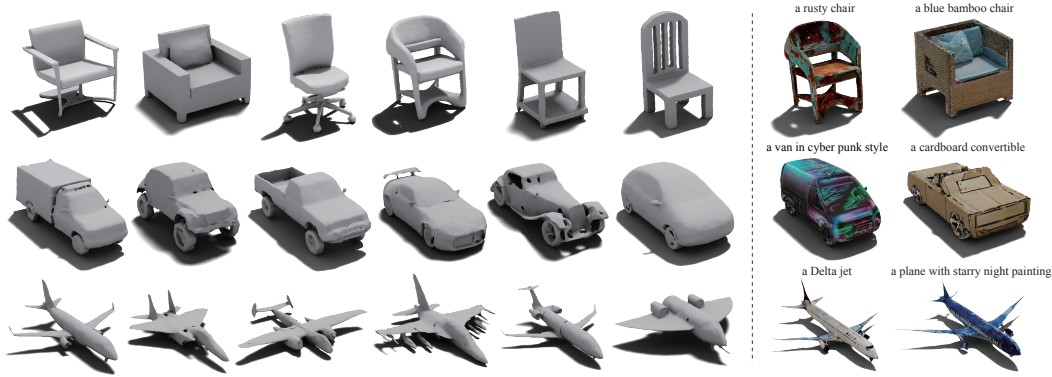

(a) Unconditionally generated 3D mesh samples from MeshDiffusion          (b) Our meshes with text-conditioned textures

Figure 1: (a) Unconditionally generated 3D mesh samples randomly selected from the proposed *MeshDiffusion*, a simple diffusion model trained on a direct parametrization of 3D meshes without bells and whistles. (b) 3D mesh samples generated by *MeshDiffusion* with text-conditioned textures from [39]. *MeshDiffusion* produces highly realistic and fine-grained geometric details while being easy and stable to train.

## ABSTRACT

We consider the task of generating realistic 3D shapes, which is useful for a variety of applications such as automatic scene generation and physical simulation. Compared to other 3D representations like voxels and point clouds, meshes are more desirable in practice, because (1) they enable easy and arbitrary manipulation of shapes for relighting and simulation, and (2) they can fully leverage the power of modern graphics pipelines which are mostly optimized for meshes. Previous scalable methods for generating meshes typically rely on sub-optimal post-processing, and they tend to produce overly-smooth or noisy surfaces without fine-grained geometric details. To overcome these shortcomings, we take advantage of the graph structure of meshes and use a simple yet very effective generative modeling method to generate 3D meshes. Specifically, we represent meshes with deformable tetrahedral grids, and then train a diffusion model on this direct parametrization. We demonstrate the effectiveness of our model on multiple generative tasks.

## 1 INTRODUCTION

As one of the most challenging tasks in computer vision and graphics, generative modeling of high-quality 3D shapes is of great significance in many applications such as virtual reality and metaverse [11]. Traditional methods for generative 3D shape modeling are usually built upon representations of voxels [51] or point clouds [1], mostly because ground truth data of these representations are relatively easy to obtain and also convenient to process. Both representations, however, do not produce fine-level surface geometry and therefore cannot be used for photorealistic rendering of shapes of different materials in different lighting conditions. And despite being convenient to process

---
*Work done partially during an internship at Max Planck Institute for Intelligent Systems.

for computers, both voxels and point clouds are relatively hard for artists to edit, especially when the generated 3D shapes are complex and of low quality. Moreover, modern graphics pipelines are built and optimized for explicit geometry representations like meshes, making meshes one of the most desirable final 3D shape representations. While it is still possible to use methods like Poisson reconstruction to obtain surfaces from voxels and points clouds, the resulted surfaces are generally noisy and contain many topological artifacts, even with carefully tuned hyperparameters.

To improve the representation flexibility, sign distance fields (SDFs) have been adopted to model shape surfaces, which enables us to use marching cubes [29] to extract the zero-surfaces and thus 3D meshes. However, SDFs are typically harder to learn as it requires a carefully designed sampling strategy and regularization. Because SDFs are usually parameterized with multi-layer perceptrons (MLPs) in which a smoothness prior is implicitly embedded, the generated shapes tend to be so smooth that sharp edges and important (and potentially semantic) details are lost. Moreover, SDFs are costly to render and therefore less suitable for downstream tasks like conditional generation with RGB images, which require an efficient differentiable renderer during inference.

We instead aim to generate 3D shapes by directly producing 3D meshes, where surfaces are represented as a graph of triangular or polygon faces. With 3D meshes, all local surface information is completely included in the mesh vertices (along with the vertex connectivity), because the surface normal of any point on the shape surface is simply a nearest neighbor or some local linear combination of vertex normals. Such a regular structure with rich geometric details enables us to better model the data distribution and learn generative models that are more geometry-aware. In light of recent advances in score-based generative modeling [16, 47] where powerful generative performance and effortless training are demonstrated, we propose to train diffusion models on these vertices to generate meshes. However, it is by no means a trivial task and poses two critical problems: (1) the numbers of vertices and faces are indefinite for general object categories, and (2) the underlying topology varies wildly and edges have to be generated at the same time.

A natural solution is to enclose meshes with another structure such that the space of mesh topology and spatial configuration is constrained. One common approach is to discretize the 3D space and encapsulate each mesh in a tiny cell, and it is proven useful in simulation [37] and human surface modeling [35]. Observing that this sort of mesh modeling is viable in recent differentiable geometry modeling literature [32, 38, 43], we propose to train diffusion models on a discretized and uniform tetrahedral grid structure which parameterizes a small yet representative family of meshes. With such a grid representation, topological change is subsumed into the SDF values and the inputs to the diffusion model now assume a fixed and identical size. More importantly, since SDF values are now independent scalars instead of the outputs of an MLP, the parameterized shapes are no longer biased towards smooth surfaces. Indeed, by such an explicit probabilistic modeling we introduce an explicit geometric prior into shape generation, because a score matching loss of diffusion models on the grid vertices has direct and simple correspondence to the vertex positions of triangular mesh.

We demonstrate that our method, dubbed *MeshDiffusion*, is able to produce high-quality meshes and enables conditional generation with a differentiable renderer. MeshDiffusion is also very stable to train without bells and whistles. We validate the superiority of the visual quality of our generated samples qualitatively with different rendered views and quantitatively by proxy metrics. We further conduct ablation studies to show that our design choices are necessary and well suited for the task of 3D mesh generation. Our contributions are summarized below:

- To our knowledge, we are the first to apply diffusion model for unconditionally generating 3D high-quality meshes and to show that diffusion models are well suited for 3D geometry.
- Taking advantage of the deformable tetrahedral grid parametrization of 3D mesh shapes, we propose a simple and effortless way to train a diffusion model to generate 3D meshes.
- We qualitatively and quantitatively demonstrate the superiority of MeshDiffusion on different tasks, including (1) unconditional generation, (2) conditional generation and (3) interpolation.

## 2 RELATED WORK

**3D Shape Generation**. 3D shape generation is commonly done by using generative models on voxels [51] and point clouds [53], due to their simplicity and accessibility. These resulted (often noisy) voxels or point clouds, however, do not explicitly encode surface information and therefore has to be processed with surface reconstruction methods in order to be used in applications like relighting and

simulation. Recent advances in implicit representations [31] lead to a series of generative models on neural fields [5, 6, 23, 42]. Based on these representations, one can learn 3D shapes directly from 2D images [5, 6, 42] with differentiable rendering in the generator. Most of these implicit representations, despite representing shapes in a more photorealistic way, require additional post-processing steps to extract explicit meshes, and are often more time-consuming to render images. If the underlying topology is known, one may directly generate the vertices of a mesh [30, 48]. [33] extends this approach to the topology-varying case with autoregressive models, sequentially generating vertices and edges, but it is hardly scalable and yields unsatisfactory results on complex geometry. A batch of concurrent work propose similar solutions to mesh generation, including: GET3D [13] which uses StyleGAN [19] with a differentiable renderer on tetrahedral grid representations and learns to generate 3D meshes from 2D RGB images, TetGAN [14] which trains generative adversarial networks (GANs) on tetrahedral grids and LION [54] which uses a trained Shape-As-Points [36] network to build meshes from diffusion-model-generated latent point clouds.

**Mesh Reconstruction**. Reconstructing 3D meshes of generic objects is challenging and often ill-posed due to its highly complex structures [18, 20, 28, 50, 52]. One often resorts to some non-mesh intermediate representations that are easy to process, and then transforms them back to meshes with mesh reconstruction methods. One of the most popular methods is marching cubes [29] which assumes that a surface is represented by the zero level set of some continuous field, and this continuous field can be well approximated by linear interpolation of discrete grid points. It is possible to avoid implicit representations and directly construct meshes. For instance, by assuming the points lie on a surface, one can build triangular meshes by connecting these points, a process known as Delaunay triangulation [3, 10, 15, 22]. It is relatively rare because of its strong assumption on data. With known mesh topology, one can deform a mesh template to fit a given representation. This approach can be easily used to fit multiview images of shapes with known topology (*e.g.*, human faces) using a differentiable renderer [25]. It is recently shown that we are able to parametrize meshes of varying topology [38, 43] and optimize them using differentiable renderers [32], which our work leverages.

**Score-based Generative Models**. Recent years have witnessed a surge in modeling data distributions with score-based models [16, 46, 47], which parameterizes the logarithm of the gradient of the probability, known as the score function, rather than the probability directly. Different from energy-based models, the often intractable normalization constant can be avoided and therefore training can be performed by simply matching the score function. It has been shown that, by using multi-scale models [16], U-Net architectures [41] and denoising score matching [49], score-based models can perform high-fidelity image generation and inpainting [16, 40, 47].

## 3 PRELIMINARIES

**Deep marching tetrahedra** (DMTet) [43] is a method to parametrize and optimize meshes of arbitrary topology in a differentiable way. The 3D space is discretized with a deformable tetrahedral grid, in which each vertex possesses a SDF value. The SDF of each 3D position in the space is computed by marching tetrahedra [12], which assumes SDF values to be barycentric interpolation of the SDF values of the vertices of the enclosing tetrahedra. More specifically, for a point $x_q$ inside a tetrahedron of vertices $x_1, x_2, x_3, x_4$ with SDF values being $s_1, s_2, s_3, s_4$, we obtain its unique barycentric coordinate $(a_1, a_2, a_3, a_4)$ such that $x_q = \sum_i a_i x_i$ ($a_i \in [0, 1]$). The SDF value $s_q$ of $x_q$ is then computed as $s_q = \sum_i a_i s_i$. As a result, if there exists a triangular mesh in the tetrahedron (*i.e.*, $s_1, s_2, s_3, s_4$ are not of the same sign), then we can know that the triangular mesh is exactly the zero surface in the tetrahedron. The triangular mesh vertex $v_p$ on a tetrahedron edge $(v_a, v_b)$ is therefore computed by $v_p = (v_a s_b - v_b s_a)/(s_b - s_a)$. The advantage of using DMTet, compared to variants of deep marching cubes [26, 38] which rely on the marching cubes algorithm [29], is that the grids are deformable and thus more capable of capturing some finer geometric details like very thin handles of chairs. While the cubic grid used for deep marching cubes can be deformable as well, the deformed cubes are much worse objects to deal with, compared to tetrahedra which will remain tetrahedra after deformation. Notably, DMTet can be fitted with a differentiable renderer by jointly optimizing for the geometry, the texture and the lighting of a 3D shape given multi-view RGB images within a reasonable amount of time and memory budget [32].

**Diffusion models** are a family of score-based generative models which learn and infer with the ground-truth time-dependent score function defined by a forward diffusion process [16]. Under the stochastic differential equation (SDE) framework proposed in [47], diffusion models construct the probability of each point in a space by diffusing all data points $x_0 \in \mathcal{D}$ to a target distribution at time $T$

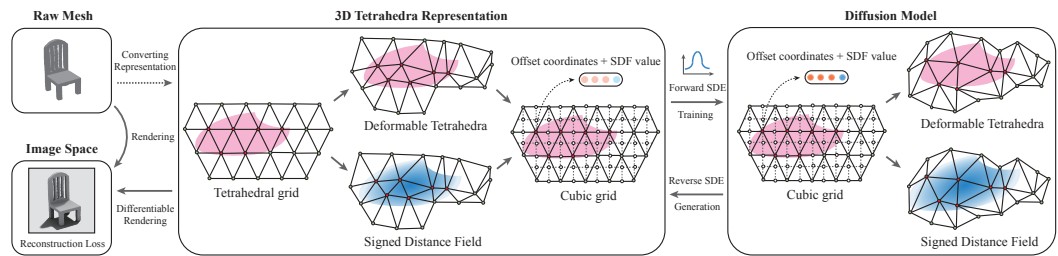

Figure 2: Overview of the proposed MeshDiffusion model.

(normally $p_T(x_T) = \mathcal{N}(x_T; 0, I)$) in a time-variant manner: $\log p_t(x_t|\mathcal{D}) = \mathbb{E}_{x_0 \in \mathcal{D}}[\log p_t(x_t|x_0)]$, where $p_t(x_t|x_0)$ is modeled by the following Itô process, with $w(t)$ being a standard Wiener process: $dx = F(x, t)dt + G(t)dw$, in which $F(x, t)$ and $G(t)$ are some predefined functions. The training objective is to approximate the score function $\nabla_x \log p_t(x_t|x_0)$ with a neural network denoted by $s_\theta(x_t, t)$, where $\lambda(t)$ is a scalar weight subject to the model design:

$$L(\theta; \mathcal{D}) = \mathop{\mathbb{E}}_{t \in [T], x_0 \in \mathcal{D}} \lambda(t) \|s_\theta(x_t, t) - \nabla_x \log p_t(x_t|x_0)\|_2^2. \tag{1}$$

With the trained score function predictor, samples can be generated by solving the reverse SDE [47] with the initial distribution $p_T(x_T)$ using solvers like Euler-Maruyama method [21]:

$$dx = \big(F(x, t) - G^2(x)\nabla_x \log p_t(x_t|x_0)\big)dt + G(t)dw. \tag{2}$$

A popular diffusion model DDPM [16], which utilizes a discrete diffusion process $p(x_t|x_{t-1}) = \mathcal{N}(x_t; (1 - \beta_t)x_{t-1}, \beta_t I)$, can be formulated within this SDE framework as [47]: $dx = -\beta_t x dt + \sqrt{\beta_t}dw$. The parameters $\beta_t$ are chosen such that $p_T(x_T)$ is close to a standard Gaussian distribution.

## 4 *MeshDiffusion*: DIFFUSION MODEL ON MESHES

With DMTet, 3D meshes can be approximated and parameterized with tetrahedral grids. Moreover, by initializing the tetrahedral grid using body-centered cubic (BCC) tiling [24] and treating the deformation of each vertex only as an attribute, we obtain a uniform and almost-regular tetrahedral grid (with fewer vertex degrees on the boundary) in which 3D translational symmetry is preserved. Such nice properties enable us to easily train a diffusion model for generating 3D meshes. From now on, we always assume that the undeformed tetrahedral grid is *uniformly* initialized and predict the deformation of tetrahedral grid vertices from their initial positions. As a result, our model takes as input a uniform tetrahedral grid with 4-dimensional attributes (specifically, 3 dimensions are for deformation and 1 dimension is for SDF). These structural priors inspire us to use 3D convolutions.

### 4.1 3D CONVOLUTIONAL NETWORKS FOR DEFORMABLE TETRAHEDRAL GRIDS

While it is natural to use graph neural networks (GNNs) for diffusion models on tetrahedral grids, we argue that it is better to use convolutional neural networks (CNNs) which generally have better model capacity and contextual information than GNNs due to the embedded spatial priors in the convolution operators. Specifically, the set of vertex positions of a *uniform* tetrahedral grid, created in the way described in [24], is a subset of the vertex position set of a uniform cubic lattice. A standard voxel-like representation can then be easily created by infilling the "missing" sites of a tetrahedral grid (illustrated in Figure 2). We follow the common practices in diffusion models [47] and use a 3D U-Net for the score function predictor. Detailed architectures and settings are in the appendix.

### 4.2 TRAINING OBJECTIVE

Since the tetrahedral grid representation of each object in the datasets is not given, our training objective can be formulated by the following constrained optimization:

$$\min_\theta \mathop{\mathbb{E}}_{\substack{t \in \text{Cat}(\{1, ..., T\}), y_0 \in \mathcal{D} \\ x_t \sim p(x_t|x_0 = g_{\phi^*}(y_0))}} \lambda(t) \|s_\theta(x_t, t) - \nabla_x \log p_t(x_t|g_{\phi^*}(y_0))\|_2^2 \tag{3}$$

$$\text{s.t.} \quad \phi^* = \arg\min_\phi L_{\text{Render}}\big(y_0, g_\phi(y_0)\big),$$

in which $\mathcal{D}$ is the 3D shape dataset, $y_0 \sim \mathcal{D}$ is the set of 2D views of an object sampled from $\mathcal{D}$, $t$ is uniformly sampled from $\{1, ..., T\}$, $s_\theta(x, t)$ is the score function approximator parameterized by a

| **Algorithm 1** Training and Inference | **Algorithm 2** Conditional Generation |
|---|---|
| **Training:** | |
| 1: Fitting $x'$ for each $y \in \mathcal{D}$, and normalize the SDF values to $\pm 1$, resulting the dataset $(x', y) \in \mathcal{D}'$. | 1: Randomly initialize and fit a deformable tetrahedral grid using the given RGBD view. |
| 2: Fitting $x$ for each $(x', y) \in \mathcal{D}'$ by conditioning the SDF values on those of $x'$. | 2: Using rasterization, find all occluded tetrahedra of which we mask all tetrahedral vertices. |
| 3: Train a diffusion model $s_\theta(x, t)$ on $\mathcal{D}'$ by treating the normalized SDF values as float numbers. | 3: Re-initialize the masked tetrahedral vertices and run the completion algorithm using the trained diffusion model. |
| **Inference:** | |
| 1: Obtain $x_0$ by solving the reverse SDE with initial distribution $q(x_T)$ and normalize the SDF values of $x_0$ to $\pm 1$. | 4: (Optionally) Finetune the unmasked tetrahedral vertices as well near the end of the completion process. |
| 2: (Optionally) Regenerate $x_0$ by conditioning on the previously generated (and normalized) SDF values. | |

neural network $\theta$, $g_\phi(y_0)$ is the mapping from 2D views of $y_0$ to its tetrahedral grid representation and $L_{\text{Render}}$ is the rendering (and relevant regularization) loss used in [32].

While in theory it is possible to train $g_\phi$ such that $s_\theta$ can be better learned, we instead take the simplest two-stage and non-amortized approach as detailed optimization and encoder architecture design to use is beyond our scope. Specifically, we first solve for the constraint $L_{\text{Render}}$ and create a dataset $\mathcal{D}_x$ of tetrahedral grids by fitting a tetrahedral grid $x_0$ for each $y_0 \sim \mathcal{D}$ in a non-amortized way. The second stage is simply to optimize for the main objective, *i.e.*, a $L_2$ denoising loss of the diffusion model, on this generated dataset $\mathcal{D}_x$. Despite simplicity, we find that this simple procedure is very stable and yields reasonably good performance.

In principle, the first data fitting stage can be trained with multiview RGB images by following the procedure in [32]. However, we notice that with RGB images only, it fails to learn some complex geometries, especially when surface materials are highly specular. To demonstrate our idea while not delving too deep into the rendering details, we instead assume that we have access to 3D mesh datasets (in our experiments, ShapeNet datasets [7]) so that RGBD images can be rendered with random lighting and simple surface materials. More specifically, we use a single default material with diffuse components only for all ground truth meshes, and render multiview RGBD images with some known but randomly rotated environment light (represented as a cubemap [44]). With the additional depth information, our reconstruction objective is $L_{\text{Render}} = \alpha_{\text{image}}L_{\text{image}} + \alpha_{\text{depth}}L_{\text{depth}} + \alpha_{\text{chamfer}}L_{\text{chamfer}} + \alpha_{\text{penalty}}L_{\text{penalty}}$, in which $L_{\text{image}}$ and $L_{\text{depth}}$ are the image loss (RGB and silhouette) and the depth loss between the rendered views and the ground truth views, respectively; $L_{\text{chamfer}}$ is the Chamfer distance [1] between a sampled point cloud from the predicted mesh and one from the ground truth mesh; $L_{\text{penalty}}$ is the regularization terms to move reconstructed meshes out from bad local minima. Details of these losses are included in the appendix.

In order to remove absolute scales from SDFs (Section 4.3), we perform the optimization twice, where during the second pass we fix the SDFs to be the signs (*i.e.*, $\pm 1$) of the resulted SDF values from the first pass. This can effectively improve the results.

## 4.3 REDUCING NOISE EFFECT OF MARCHING TETRAHEDRA

It is tempting to train diffusion models on arbitrary $\mathcal{D}_x$, the dataset of fitted tetrahedral grids of objects, with low $L_{\text{Render}}$. However, the naively trained models generate highly non-smooth surfaces (Figure 3) due to a mismatch between the $L_2$ denoising objective on the 4-dimensional inputs (deformation and SDF) and the ideal $L_2$ reconstruction objective on triangular mesh vertex positions. Recall that the triangular mesh vertex position $v_p$ on a single tetrahedron edge $e = (a, b)$ is computed by linear interpolation $v_p = (v_a s_b - v_b s_a)/(s_b - s_a)$, in which $v_a, v_b$ are the positions of tetrahedron vertices $a$ and $b$, and $s_a, s_b$ are the corresponding SDF values. With perfectly fitted $v_a$ and $v_b$ but a small noise $\epsilon$ on both $s_a$ and $s_b$, the triangular mesh vertex is subject to a perturbation inversely proportional to $s_b - s_a$, which may incur unexpected consequences: $v_{p,\text{noisy}} - v_p = \epsilon(v_a - v_b)/(s_b - s_a)$. An arbitrarily fitted dataset $\mathcal{D}_x$ does not provide any guarantee on the scale of $|s_b - s_a|$ across different locations in different data points. Such a behavior under the simplified assumption implies that the error in predicted mesh vertex positions can vary a lot even with the same $L_2$ denoising loss in SDFs and deformation, indicating unevenly weighted training and inference in the view of mesh vertices.

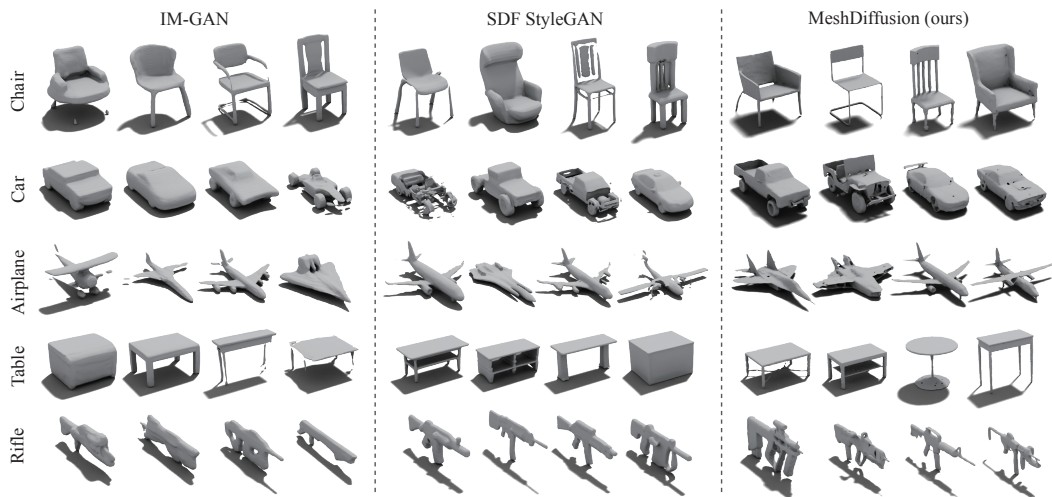

Figure 4: Qualitative comparison among different generative models. All the samples are randomly selected.

A similar issue arises when we consider the topological changes due to a small noise in the SDF values. For illustration, we consider the case where some tetrahedral vertex $s_a$ with $-C < s_a < 0$ ($C \ll 1$ is a tiny positive scalar), and all its neighboring vertices possess SDF value of 1. As a result, with a small noise on $s_a$, the probability of a local topological change (*e.g.*, an undesirable hole) in the resulted 3D meshes can differ a lot even when the $L_2$ denoising loss is the same.

Due to these undesired behaviors, we propose to normalize the SDF values and adopt the two-pass optimization scheme described in Section 4.2, such that we can ensure $s_a, s_b \in \{+1, -1\}$ and $|s_b - s_a| = 2$ for all mesh-generating tetrahedron edges.

With normalized SDFs values and continuous deformation vectors, it is natural train a hybrid diffusion model in which the SDFs are modeled like in D3PM [2] since normalized SDFs take discrete values of $\pm 1$. However, we empirically find that it suffices to simply treat the normalized SDFs as float numbers and train a standard Gaussian diffusion model. The inference thus requires a final normalization operation to round SDFs to $\pm 1$ according to their signs.

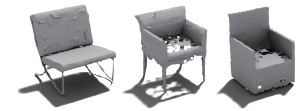

Figure 3: Undesirable artifacts produced by naïve DDPM due to the sensitivity of marching tetrahedra to SDF noises. Better viewed by zooming in.

Because the generative model is trained under the assumption that SDF values are un-normalized, a further refinement step can be applied to improve visual quality. Specifically, we simply set the generated and normalized SDFs as the conditional inputs and then run a conditional generation step on the deformation vectors only.

### 4.4 Conditional Generation with Single-view Images

In many cases, we would like to generate 3D objects given a (possibly partial) single view RGBD image. Such a task is simple with MeshDiffusion by a two-stage process: first to fit a tetrahedral grid with the given single RGBD view and second to use our MeshDiffusion model to (stochastically) correct the wrong parts. We follow the procedure shown in Algorithm 2. For the conditional inference process, we consider the simplest conditional generation method [16] (also called replacement method in [17]). Specifically, we sample $\hat{x}_{t-1}$ from $x_t$ and replace $\hat{x}_{t-1}^b$ by $p_{t-1}(x_{t-1}^b|x_0^a)$ which is defined by the forward diffusion. More details are included in the appendix.

## 5 Experiments and Results

### 5.1 General Settings

**Data Fitting**. We fit tetrahedral grids on five ShapeNet subcategories: Chair, Airplane, Car, Rifle and Table, the same set of categories used in [9, 55]. Fitting each object takes roughly 20-30 minutes on a single Quadro RTX 6000 GPU. We use the same train/test split in [55]. The detailed architecture, training and hyperparameter settings are explained in the appendix.

| Method | | MMD (↓) | | | COV (%, ↑) | | | 1-NNA (%, ↓) | | | JSD ($10^{-3}$, ↓) |
|---|---|---|---|---|---|---|---|---|---|---|---|
| | | CD | EMD | LFD | CD | EMD | LFD | CD | EMD | LFD | |
| Chair | IM-GAN | 13.928 | 1.816 | 3615 | **49.64** | 41.96 | **47.79** | 58.59 | 69.05 | 68.58 | 6.298 |
| | SDF-StyleGAN | 15.763 | 1.839 | 3730 | 45.60 | 45.50 | 43.95 | 63.25 | 67.80 | 67.66 | 6.846 |
| | GET3D | 15.972 | 1.843 | 3801 | 43.36 | 42.77 | 44.48 | 75.26 | 72.49 | 82.82 | **4.732** |
| | MeshDiffusion | **13.212** | **1.731** | **3472** | 46.00 | **46.71** | 42.11 | **53.69** | **57.63** | **63.02** | 5.038 |
| Car | IM-GAN | 5.209 | 1.197 | 2645 | 28.26 | 24.92 | 30.73 | 95.69 | 94.79 | 89.30 | 42.586 |
| | SDF StyleGAN | 5.064 | **1.152** | 2623 | 29.93 | **32.06** | 41.93 | 88.34 | 88.31 | 84.13 | 15.960 |
| | GET3D | 6.243 | 1.252 | 2657 | 15.04 | 18.38 | 31.13 | **75.26** | **72.49** | 89.07 | 69.107 |
| | MeshDiffusion | **4.972** | 1.196 | **2477** | 34.07 | 25.85 | **37.53** | 81.43 | 87.84 | **70.83** | 12.384 |
| Airplane | IM-GAN | 3.736 | 1.110 | 4939 | 44.25 | 37.08 | **45.86** | 79.48 | 82.94 | 79.11 | 21.151 |
| | SDF StyleGAN | 4.558 | 1.180 | 5326 | 40.67 | 32.63 | 38.20 | 85.48 | 87.08 | 84.73 | 26.304 |
| | MeshDiffusion | **3.612** | **1.042** | **4538** | **47.34** | **42.15** | 45.36 | **66.44** | **76.26** | **67.24** | **11.366** |
| Rifle | IM-GAN | 3.550 | 1.058 | 6240 | 46.53 | 37.89 | 42.32 | 70.00 | 72.74 | 69.26 | 25.704 |
| | SDF StyleGAN | 4.100 | 1.069 | 6475 | 46.53 | 40.21 | 41.47 | 73.68 | 73.16 | 76.84 | 33.624 |
| | MeshDiffusion | **3.124** | **1.018** | **5951** | **52.63** | **42.11** | **48.84** | **57.68** | **67.79** | **55.58** | **19.353** |
| Table | IM-GAN | **11.378** | 1.567 | 3400 | **51.04** | 49.20 | 51.04 | 65.96 | 63.17 | 62.49 | 4.865 |
| | SDF StyleGAN | 13.896 | 1.615 | **3423** | 42.21 | 41.80 | 42.98 | 68.35 | 68.21 | 66.19 | 4.603 |
| | MeshDiffusion | 11.405 | **1.548** | 3427 | 49.56 | **50.33** | **51.92** | **59.35** | **59.47** | **58.97** | **4.310** |

Table 1: Shape metrics of our model and baseline models.

**Generative Model**. To show the effectiveness and universality of our approach of training diffusion models on 3D meshes, for the U-Net architecture we take an extremely approach by simply replacing the 2D convolution and other 2D modules in the U-Net architecture used by [47] with their 3D counterparts. To improve model capacity, we provide to the model as a conditional input a mask indicating which lattice sites are artificially introduced. We slightly increase the depth of the network due to the larger input size. We do not tune the hyperparameters of diffusion SDEs but instead use the same set of hyperparameters as described in [47], which also validates our MeshDiffusion is easy to train. We train the discrete-time category-specific diffusion models for all datasets for total 90k iterations with batch size 48 on 8 A100-80GB GPUs. The training process typically takes 2 to 3 days.

For ablation on alternative architectures, we train GANs on our datasets with an architecture similar to the one used in [51]. Our SDF-based baselines include IM-GAN [9] and SDF-StyleGAN [55]. We also compare MeshDiffusion against GET3D [13] which also uses DMTet for mesh parametrization.

## 5.2 UNCONDITIONAL GENERATION

For qualitative comparison, we randomly sample 3D shapes from each of the trained generative models for each dataset. We remove isolated meshes of tiny sizes and then apply both remeshing and the standard Laplace smoothing [34] on all the generated meshes (smoothed with $\lambda = 0.25$ and 5 optimization steps). We visualize samples produced by MeshDiffusion and the existing state-of-the-art 3D mesh generative models in Figure 4. We note that MeshDiffusion produces the sharpest samples and preserve the finest geometric details, while pure SDF-based methods tend to be too smooth. Part of the reason is that these SDF-based methods assume a very smooth interpolation between points, while MeshDiffusion explicitly models the interpolation in a piecewise linear way.

### 5.2.1 QUANTITATIVE EVALUATION

In Table 1, we show various point cloud distances [1, 55] and light field distance (LFD) [8] between the test set and the generated samples by sampling point clouds of size 2048 from both ground truth and generated meshes. The results consistently show that MeshDiffusion can better capture the geometric details, and therefore achieves better point cloud metrics in most of the cases. For a more detailed

| Model | Chair | Airplane | Car | Rifle | Table |
|---|---|---|---|---|---|
| IM-GAN [9] | 64.19 | 74.57 | 141.2 | 103.3 | 51.70 |
| SDF-StyleGAN [55] | **36.48** | 65.77 | **128.70** | 65.50 | **42.29** |
| MeshDiffusion | 39.62 | **64.30** | 130.20 | **54.73** | 48.55 |

Table 2: FID scores averaged across 24 views.

description on the metrics and experimental settings, please refer to the appendix.

We also follow [55] and measure Frechet inception distances (FIDs) [4] on rendered views as a proxy of goodness of surfaces. Same as in [55], the views are rendered with 24 camera poses uniformly distributed on a sphere; 3 fixed point light sources are used to light the meshes; a gray diffuse material is used for all meshes. The FID score of each view is computed with an ImageNet-pretrained model

| Method | MMD (↓) | | | COV (%, ↑) | | | 1-NNA (%, ↓) | | | JSD ($10^{-3}$, ↓) |
|---|---|---|---|---|---|---|---|---|---|---|
| | CD | EMD | LFD | CD | EMD | LFD | CD | EMD | LFD | |
| GAN on Tets | 16.116 | 1.537 | 4173 | 45.13 | 46.39 | 41.74 | 72.97 | 72.57 | 85.91 | 5.353 |
| MeshDiffusion | **13.212** | **1.731** | 3472 | **46.00** | **46.71** | 42.11 | **53.69** | **57.63** | 63.02 | 5.038 |
| Ours w/o Smoothing | 13.885 | 1.772 | **3446** | 43.36 | 45.50 | 43.36 | 60.88 | 62.54 | **62.32** | **4.716** |
| Ours w/o Normalization | 14.324 | 1.816 | 3690 | 44.76 | 46.61 | **44.69** | 63.94 | 65.01 | 65.34 | 5.178 |

Table 3: Ablation study of MeshDiffusion on the Chair category in ShapeNet.

and averaged to obtain the FID scores for 3D shapes. With suitable hyperparameters for resolution and Laplacian smoothing, our MeshDiffusion is able to achieve competitive scores compared to SDF-StyleGAN. We note, however, the computed FID is not a genuine score for 3D shapes since the distribution of rendered views of these meshes is very different from the distribution of natural RGB images. In addition, the multiview FID score used by SDF-StyleGAN [55] assumes flat shading, while it is more natural and common to use other shading methods like Phone shading for smoother images. Results in Table 2 show that even with such a over-simplified shading, MeshDiffusion still yields very competitive FID scores compared to recent state-of-the-art methods.

Additionally, we perform ablation study on the choices of models in Table 3. It can be observed that the our SDF normalization strategy described in Section 4.3 is indeed beneficial for the diffusion model, and our customized diffusion model is better suited for our mesh generation setting.

## 5.3 Conditional Generation

Our model can generate meshes by conditioning on a single-view RGBD image, as shown in Figure 5. Because the geometry estimated from the single-view fitting is not perfect even in the given single view, we allow the originally-fixed tetrahedral vertices to be slightly updated by the diffusion model near the end of the diffusion completion process (in our experiments, $T = 50$; the inference process starts at $T = 1000$ and ends at $T = 0$). Results demonstrate that MeshDiffusion can generate plausible and reasonable completion results conditioned on the given view.

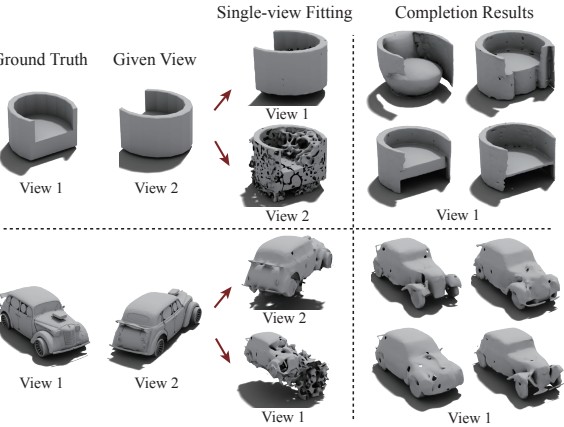

Figure 5: Conditional generation on a single RGBD view.

## 5.4 Interpolation Results

We use DDIM [45] to convert the stochastic sampling process into a deterministic one. With DDIM, the initial random Gaussian noise $x_T$ is treated as the "latent code" of the generated image, and we run the following sampling process: $x_{t-1} = \frac{\alpha_{t-1}}{\alpha_t}[x_t - (1 - \alpha_t)s_\theta(x_t, t)] + (1 - \alpha_{t-1})s_\theta(x_t, t)$. Following the settings in DDIM paper [45], we use spherical interpolation for the latent codes. We visualize some of the interpolation sequences in Figure 7. We set the number of inference steps to 100 for faster inference and use quadratic time spacing as described in [45].

## 5.5 Text-conditioned Texture Generation

We show in Figure 1 and Figure 6 that the 3D meshes generated by MeshDiffusion can be easily painted with some texture generation methods. In our experiment, we use a recent work – TEXTure [39] to generate text-conditioned textures on generated raw 3D meshes. As can be observed from the results, MeshDiffusion along with TEXTure can produce fairly realistic and reasonable textured 3D mesh models. With more advanced texture generation methods, we believe that the advantage of MeshDiffusion's high-quality 3D geometry can be better demonstrated.

## 6 Discussions

**Optimization issues with DMTet.** While DMTet is capable of fitting geometries, it fails in cases where the underlying topology is complex. Besides, it is not encouraged to learn the true topology of

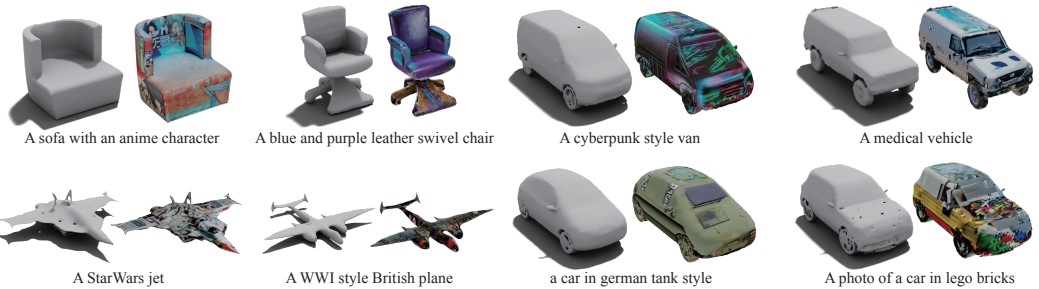

A sofa with an anime character    A blue and purple leather swivel chair    A cyberpunk style van    A medical vehicle

A StarWars jet    A WWI style British plane    a car in german tank style    A photo of a car in lego bricks

Figure 6: More examples of text-conditioned textures synthesized by [39] on our generated meshes.

shapes and may produce invisible topological holes by contracting the neighboring triangular mesh vertices close enough. Furthermore, we observe that the optimization process with differentiable rasterization of 3D meshes may produce floating and isolated meshes, especially when depth supervision is introduced. It is therefore worth designing better optimization techniques, regularization methods and possibly better parametrization of meshes for the purpose of training mesh diffusion models.

**Diffusion model design.** Our experiments demonstrate the effectiveness of the simple design with few hyperparameter and architecture changes: a 3D-CNN-based U-Net on augmented cubic grids with DDPM. We note that it is straightforward to utilize advanced diffusion models and switch to more memory-efficient architectures. Especially, since it is possible to regularize SDFs with methods such as Lipschitz constraints [27], it is a promising approach to train diffusion models on the latent space produced by a regularized autoencoder, the same strategy adopted in [40, 54].

**Limitations.** Diffusion model typically assumes a known dataset in the input modality (augmented tetrahedral grids in our case), but to efficiently train diffusion models on 2D images, we need a better way to amortize the costs and fully leverage the power of differentiable rendering. Our paper avoids this important aspect but instead adopts the two-stage approach of "reconstruction-then-generation". Moreover, in our formulation, the differentiable renderer is useful only during the tetrahedral grid creation process, while in principle we believe there can be ways to incorporate the differentiable render in the training and inference process of diffusion models. Finally, our diffusion model is built with a very naïve architecture, thus limiting the resolution of input tetrahedral grids, while we notice that some of the fine details cannot be fully captured with the current resolution of 64 during the dataset creation stage. With better architecture designs or adaptive resolution techniques (as in [43]), we may greatly increase the resolution and generate a more diverse set of fine-level geometric details.

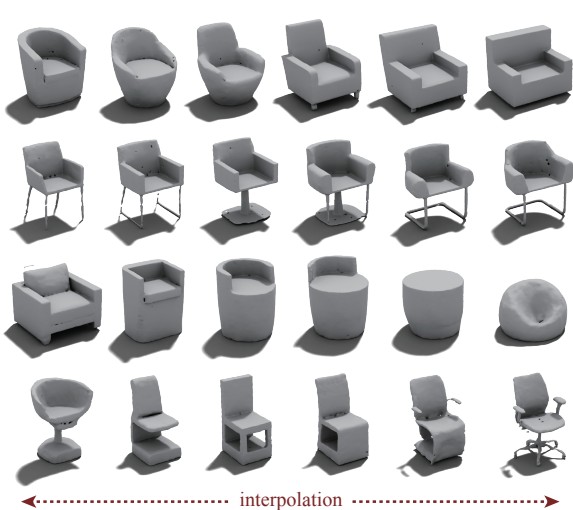

Figure 7: Some examples of our interpolation results.

# 7    CONCLUDING REMARKS

We demonstrate that our *MeshDiffusion*, a diffusion model on 3D meshes parameterized by tetrahedral grids, is able to generate fine-details of 3D shapes with arbitrary topology. Such a model is minimally designed but still outperforms the baselines. We demonstrate that, despite being trained and performing inference in the 3D space, can be easily used when only 2.5D information is available. We believe that our model can potentially shed some light on future studies of score-based models for learning and generating high-fidelity shapes. Future work may include but not limited to text-conditioned mesh generation with diffusion models, joint synthesis of texture and geometry with more realistic materials, and motion synthesis and physical simulation on generated meshes.

## ACKNOWLEDGEMENT

We thank Yuliang Xiu, Jinlong Yang, Tim Xiao, Haiwen Feng, Yandong Wen for constructive suggestions. We would like to thank Samsung Electronics Co., Ldt. for funding this research.

**Disclosure.** MJB has received research gift funds from Adobe, Intel, Nvidia, Meta/Facebook, and Amazon. MJB has financial interests in Amazon, Datagen Technologies, and Meshcapade GmbH. While MJB is a part-time employee of Meshcapade, his research was performed solely at, and funded solely by, the Max Planck Society. DN is supported by NSERC Discovery Grant (RGPIN-5011360) and LP is supported by NSERC Discovery Grant (RGPIN-04653).

**Ethics Statement.** Our model is developed for general 3D mesh generation and is in a very preliminary stage in terms of automatically generating product-quality meshes. Still, our model may potentially be used to generate inappropriate contents if trained on specific datasets.

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

# Appendix

## Table of Contents

# A    DETAILS ON DATASET PREPARATION

## A.1    ARCHITECTURE AND LOSSES

**RGB and silhouette loss.** By interpolating between a black background image and the binary mask (produced by rasterization to indicate the existence of rasterized meshes on each pixel), we obtain a smoothed binary mask $M$ and $M_{gt}$, for the fitted meshes and predicted meshes respectively. We then compute the silhouette loss:

$$L_{\text{silhouette}} = \|M - M_{gt}\|_2^2.$$

We compute the averaged pixel-wise log-$L_1$ loss between rendered images and ground truth images:

$$L_{\text{RGB}} = \mathop{\mathbb{E}}_{x \in \mathcal{D}, Q \in \mathcal{D}_{\text{pose}}} \log(\|I_{gt} \odot M_{gt} - \text{Render}(x, Q) \odot M\|)$$

in which $\mathcal{D}_{\text{pose}}$ is the set of random camera poses which always look at the center of the object, and $x$ is the sampled tetrahedral grid from the dataset $\mathcal{D}$.

**Depth loss.** We use a mixed loss for depth: when the depth is greater than a threshold (set to $1.0$), we use a $L_2$ loss; otherwise, we switch to $L_1$ loss as we find it producing much better surfaces.

$$L_{\text{depth}} = \mathop{\mathbb{E}}_{x \in \mathcal{D}, Q \in \mathcal{D}_{\text{pose}}} \|d_{gt} - \text{Render}_{\text{depth}}(x, Q)\|_{\text{mixed}}.$$

The depth images are computed by barycentric interpolation following rasterization, in which the background is assumed to have a large default depth of 20, which is much greater than the object depth (constrained in $[-1, 1]$).

To better remove inner artifacts and generate inner structures, we also include the depth loss on the second layer from rasterization, *i.e.*, the second triangular mesh (if any) intersected by each view ray. We weight this second layer depth loss term by $0.1$ as we observe that it can interfere with the first layer depth loss.

The complete depth loss term is thus

$$L_{\text{depth, complete}} = \mathop{\mathbb{E}}_{x \in \mathcal{D}, Q \in \mathcal{D}_{\text{pose}}} \left[ \|d_{gt} - \text{Render}_{\text{depth}}(x, Q)\|_{\text{mixed}} + 0.1 * \|d_{gt, \text{2nd}} - \text{Render}_{\text{depth,2nd}}(x, Q)\|_{\text{mixed}} \right].$$

**Chamfer loss.** At every iteration, we randomly sample $50,000$ points from both ground truth meshes and predicted meshes and compute the standard Chamfer distance:

$$L_{\text{Chamfer}} = \sum_{x \in P_{\text{fitted}}} \min_{y \in P_{gt}} \|x - y\|_2^2 + \sum_{y \in P_{gt}} \min_{x \in P_{\text{fitted}}} \|x - y\|_2^2.$$

**SDF regularization loss.** Following [32], we use the same $L_2$ regularization loss and penalize the difference of SDFs of two neighboring tetrahedral vertices (*i.e.*, $E_{tet}$ the set of all edges in the tetrahedral grid) as in [32] so that the occluded regions do not produce complex inner geometry. It is also helpful for optimizing tetrahedral grids.

$$L_{\text{SDF}} = \sum_{(u,v) \in E_{tet}} \|\text{SDF}(u) - \text{SDF}(v)\|^2.$$

**Hyperparameter setting.** We set $\alpha_{\text{image}} = \alpha_{\text{Chamfer}} = 1.0$ and $\alpha_{\text{depth}} = 100.0$. We set $\alpha_{\text{SDF}}$ to $0.2$ and use a linear decay of the scale towards $0.01$. We use an Adam optimizer for all the parameters with a learning rate of $5e - 4$ and $(\beta_1, \beta_2) = (0.9, 0.999)$. We train both reconstruction passes with 5000 iterations.

## A.2 IMPLMENTATION DETAILS

The initial tetrahedral grid is initialized in a cube $[-1, 1]^3$ by a dense body-centered cubic (BCC) tiling of tetrahedra [24] (see https://github.com/crawforddoran/quartet for code examples). As some tetrahedral generation packages produce additional tetrahedral vertices on the boundary of the cube which breaks translational symmetry (hence detrimental to the performance of 3D CNN), we remove such symmetry-breaking boundaries if any.

We find that using the clipped deformation vectors is better than using the *tanh*-ed deformation vectors as they remove some nonlinearity. And to ensure that the deformation vectors are always differentiable, we clip the deformation vectors only after each gradient update, but not during the forward computational pass.

We set the range of deformation to be three times of that used in [43] so that the meshes can better capture details in the absence of SDF scales (Section 4.3), especially when the grid resolution is low. As self-intersecting meshes may appear, it is necessary to perform standard mesh processing operations including remeshing and Laplacian smoothing. For grids of higher resolution (*e.g.*, 128), we instead set the deformation to be 1.5x of hat used in [43].

Because we normalize SDFs after the first reconstruction pass, it is more desirable for the underlying shape to be captured mostly by the topology implied by SDF values, not the vertex deformation. To encourage convergence to such solutions, during the first 2000 iterations, we periodically scale the learned SDF values by $0.4$ for every $300$ iterations.

We observe that depth supervision is not enough to remove some isolated floater meshes in the space. Therefore, we cull these floaters by setting the SDF values of all tetrahedral vertices lying outside the visual hull to positive values (with the assumption that positive values represent ouside-ness). Due to constraints on computational resources, we do not perform culling in a full 3D way, but use rendered shape silhouette to determine floaters visible in the rendered views only.

We notice that some of the meshes in ShapeNet are not watertight while some others have parts which are very thin. Depth supervision on these parts can lead to more topological holes. Therefore, during rasterization we extract the depths (stored in Z-buffers) of the two closet meshes for each pixel and decide if we compute the depth loss on these pixels according to the difference in depth of these two meshes. Specially, depth supervision is not included if the difference is too small or there is only one rasterized mesh (meaning that the ray only intersects with a single mesh).

Motivated by the fact that only SDFs of the mesh-generating tetrahedra (*i.e.*, tetrahedra with vertices of different SDF signs) matter, we set the SDFs of all the non-mesh-generating tetrahedral vertices to $\pm 1$ (signs depending on if vertices inside/outside the shape) to reduce the complexity of the dataset.

For the single-view reconstruction stage in the conditional generation experiments, we observe that the lack of other views leads to many floating artifacts when the target shape consists of a bulk volume. Therefore, during the first 300 iterations, we periodically (every 10 iterations) query the nearest neighbor ground truth mesh vertices for every tetrahedral vertex, and determine if the vertex is in front of the camera and not behind the current view (by computing the nearest surface normal, the displacement from the nearest face center and the view direction from camera). If the tetrahedral vertex satisfies the condition, we set its SDF value to $+1$.

# B  DETAILS ON MESH DIFFUSION MODELS

| Input |
|---|
| **Encoder** |
| 2 × ResBlocks, 3 × 3 kernel, width 64 |
| Pooling, Stride 2 |
| 3 × ResBlocks, 3 × 3 kernel, width 64 |
| Pooling, Stride 2 |
| 3 × ResBlocks (with one attention layer in between), 3 × 3 kernel, width 128 |
| Pooling, Stride 2 |
| 3 × ResBlocks, 3 × 3 kernel, width 256 |
| Pooling, Stride 2 |
| 3 × ResBlocks, 3 × 3 kernel, width 256 |
| **FC** |
| 1 × ResBlocks, 3 × 3 kernel, width 256 |
| Attention |
| 1 × ResBlocks, 3 × 3 kernel, width 256 |
| **Decoder** |
| 3 × ResBlocks, 3 × 3 kernel, width 256 |
| Upsampling, Stride 2 |
| 3 × ResBlocks, 3 × 3 kernel, width 256 |
| Upsampling, Stride 2 |
| 3 × ResBlocks (with one attention layer in between), 3 × 3 kernel, width 128 |
| Upsampling, Stride 2 |
| 3 × ResBlocks, 3 × 3 kernel, width 64 |
| Upsampling, Stride 2 |
| 2 × ResBlocks, 3 × 3 kernel, width 64 |

| Input |
|---|
| **Encoder** |
| 2 × ResBlocks, 5 × 5 kernel, width 128 |
| Pooling, Stride 2 |
| 2 × ResBlocks, 3 × 3 kernel, width 128 |
| Pooling, Stride 2 |
| 2 × ResBlocks, 3 × 3 kernel, width 256 |
| Pooling, Stride 2 |
| 2 × ResBlocks (with one attention layer in between), 3 × 3 kernel, width 256 |
| Pooling, Stride 2 |
| 2 × ResBlocks, 3 × 3 kernel, width 512 |
| Pooling, Stride 2 |
| 2 × ResBlocks, 3 × 3 kernel, width 512 |
| **FC** |
| 1 × ResBlocks, 3 × 3 kernel, width 512 |
| Attention |
| 1 × ResBlocks, 3 × 3 kernel, width 512 |
| **Decoder** |
| 2 × ResBlocks, 3 × 3 kernel, width 512 |
| Upsampling, Stride 2 |
| 2 × ResBlocks, 3 × 3 kernel, width 512 |
| Upsampling, Stride 2 |
| 2 × ResBlocks (with one attention layer in between), 3 × 3 kernel, width 256 |
| Upsampling, Stride 2 |
| 2 × ResBlocks, 3 × 3 kernel, width 256 |
| Upsampling, Stride 2 |
| 2 × ResBlocks, 3 × 3 kernel, width 128 |
| Upsampling, Stride 2 |
| 2 × ResBlocks, 5 × 5 kernel, width 128 |

Table 4: Architecture of the 3D U-Net (Left: resolution 64, Right: resolution 128). The shortcuts from the encoder to the decoder are not shown.

We adapt the network of DDPM in [16] and use a base width of 64. The encoder in the U-Net is shown in Table 4, and the decoder follows the same but reverse pattern. For higher resolution grids, we slightly reduce the number of layers in each resolution stage but double the base width.

All the values at the artificially-introduced sites of a cubic lattice are set to zero. We append to the cubic grid in the first and last few layers a binary mask that indicates which vertices are from the tetrahedral grid and which are fake. The score matching loss is also masked accordingly so that the augmented vertices in the predicted cubic grids do not contribute.

To prevent overfitting, we augment the dataset by randomly translate all tetrahedron vertices by the same but tiny amount.

## C  QUANTITATIVE METRICS FOR 3D MESHES

We briefly explain the metrics we used for evaluating the quality of 3D Meshes.

**Minimum Matching Distance (MMD).** MMD measures the average distance to the nearest neighbor of individual points in one point cloud to another point cloud.

**Coverage.** Suppose every point cloud in a set $A$ is approximated by its nearest neighbor in another set $B$. Coverage measures the fraction of elements in $B$ which are used to cover $A$ in the nearest neighbor matching sense. Higher the coverage, Better that $A$ is a representative set of $B$.

**Leave-one-out Accuracy (1-NNA).** As its name indicates, 1-NNA is the average leave-one-out accuracy of the 1-NN classifier fitted on $A$ to classify $B$. It measures if each element in the set $A$ is important in representing another set $B$. A low 1-NNA score means that $A$ well covers $B$.

**Jensen-Shannon Divergence (JSD).** With JSD, we compute the distance between distribution of ground truth point clouds and that of the generated point clouds: $\text{JSD}(P_A, P_B) = \frac{1}{2}\text{KL}(P_A||M) + \frac{1}{2}\text{KL}(P_B||M)$, in which $M = \frac{1}{2}(P_A + P_B)$.

**Light Field Distance (LFD).** LFD leverages the so-called light field descriptor [8], which combines local region-based and contour-based descriptors to measure the silhouettes and thus object shapes.

# D  GENERATION SEQUENCE

For each time step $t \in \{1, ..., T\}$ during the DDPM inference process, we can compute the predicted $x_0$ by the current value of $x_t$ [16]:

$$\hat{x}_0 = \frac{1}{\sqrt{\overline{\alpha}_t}}(x_t - \sqrt{1 - \overline{\alpha}_t}\epsilon_\theta(x, t)),$$

in which $\epsilon_\theta(x, t)$ is the learned denoising network, $\alpha_t = \Pi_{s=1}^t(1 - \beta_s)$ and $\beta_t$ is the noise scale hyperparameter in DDPM.

We show example generation trajectories in Figure 8 by visualizing the predicted $x_0$ through time. All $\hat{x}_0$'s are clipped to $[0, 1]$. We visualize the sequence from $T = 250$ to $T = 50$, as the predicted tetrahedral grids before $T = 250$ produce nothing other than noise.

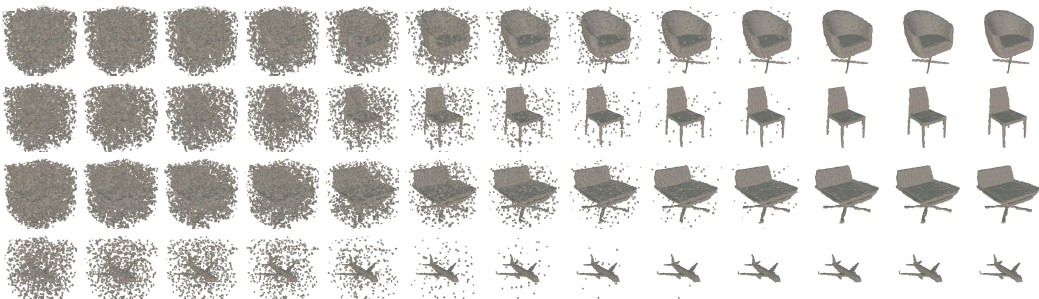

Figure 8: Generation sequences of predicted $x_0$.

# E  NEAREST NEIGHBOR VISUALIZATION

We show the nearest neighbors of some generated samples in the validation set in Figure 9.

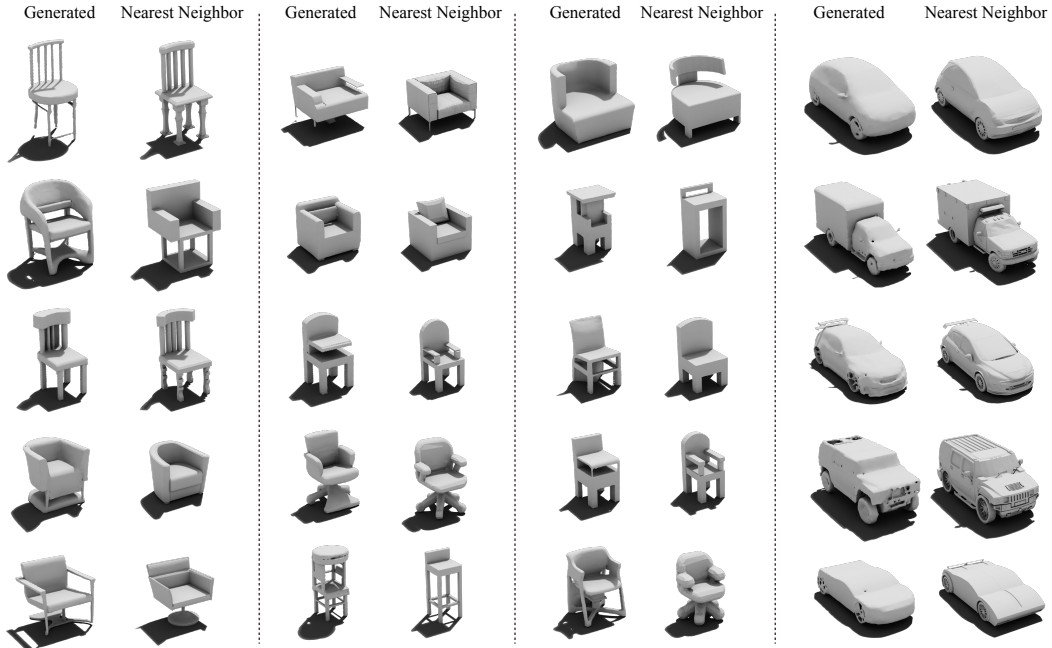

Figure 9: Nearest neighbor (from the ShapeNet dataset) of the generated samples.

## F    QUALITATIVE COMPARISON TO GET3D

We visualize some generated meshes from MeshDiffusion and GET3D in Figure 10. Both models are able to generate finer details of shapes due to the use of DMTet. On the car category of ShapeNet, generated meshes by GET3D tend to have fewer or less noticeable holes, possibly because GET3D is directly trained on adversarial losses on rendered images. In comparison, MeshDiffusion has to learn the complex and often invisible 3D structures, which is also one of our advantages for modeling complex 3D geometry with inner structures. However, potentially due to the limited capacity of the decoder (compared to a recurrent U-Net in diffusion models) and the hardness of GAN training (*v.s.* supervised learning with denoising loss), we observe that the failure cases appear more often in GET3D compared to MeshDiffusion. And the novel examples in MeshDiffusion seem to make more sense compared to those from GET3D.

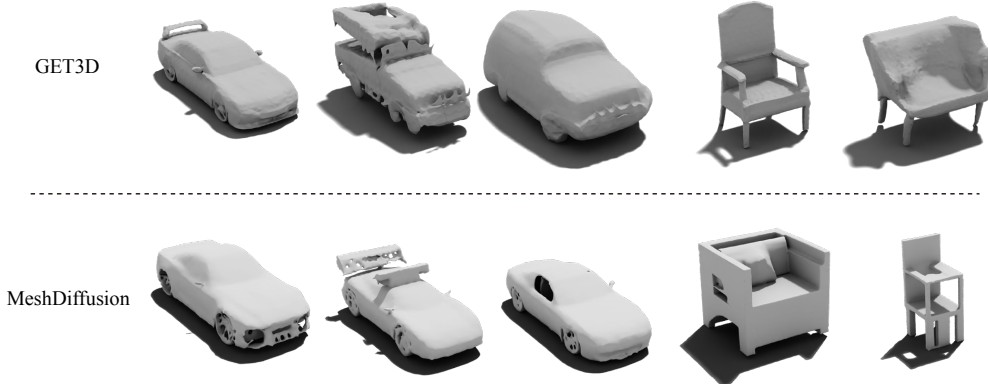

Figure 10: Qualitative Comparison between MeshDiffusion and GET3D.

# G  INNER STRUCTURE GENERATION

Since our MeshDiffusion is explicitly trained with 3D information, it is capable of generating inner structures invisible from the outside. We give a comparison between samples generated by MeshDiffusion (trained with a higher resolution of 128) and GET3D in Figure 11.

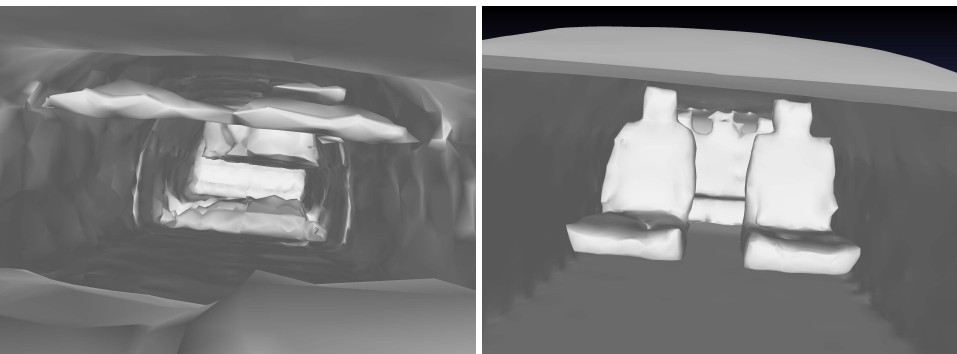

Figure 11: Inner structures of generated car meshes. Left: GET3D, Right: MeshDiffusion.

To fully evaluate the capability of surface generation (as GET3D is only trained to optimize surface appearance), we follow the same procedure described in https://github.com/nv-tlabs/GET3D/tree/master/evaluation_scripts and compare the point cloud metrics in Table 5, with generated sample set 5x larger than the validation set.

| Model | MMD (↓) | | COV (%, ↑) | |
|---|---|---|---|---|
| | CD | EMD | CD | EMD |
| GET3D | 9.8487 | 0.3668 | 42.93 | 54.20 |
| MeshDiffusion | **6.7965** | **0.3997** | **64.13** | **68.80** |

Table 5: Shape metrics of MeshDiffusion and GET3D on surface point clouds of generated car meshes.

## H MORE UNCONDITIONAL GENERATION SAMPLES

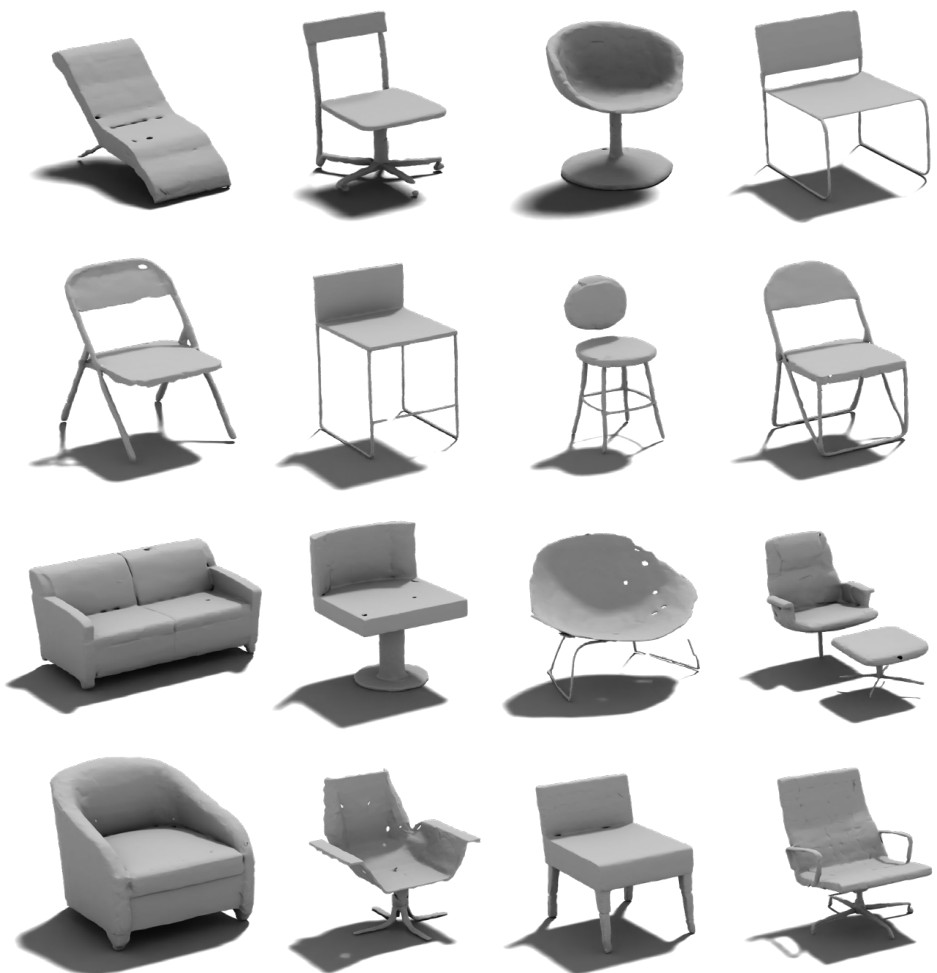

Figure 12: Generated 3D chair meshes.

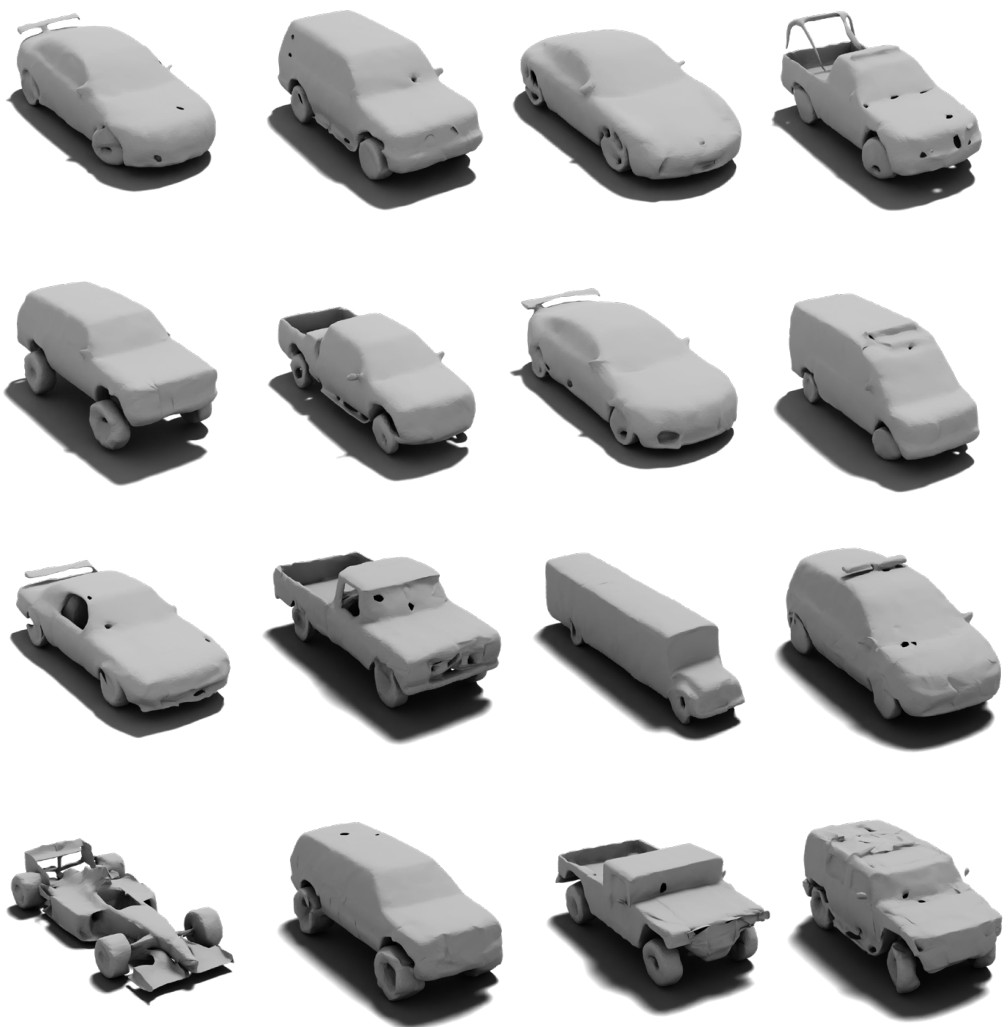

Figure 13: Generated 3D car meshes.

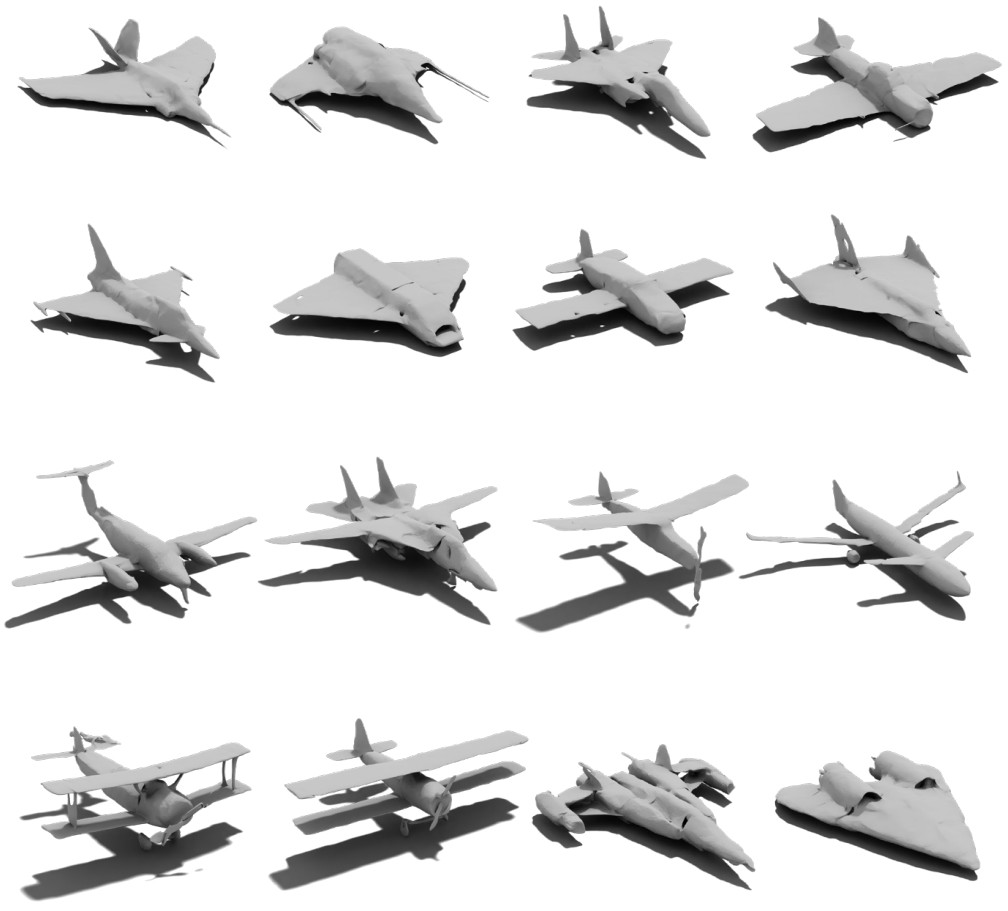

Figure 14: Generated 3D airplane meshes.

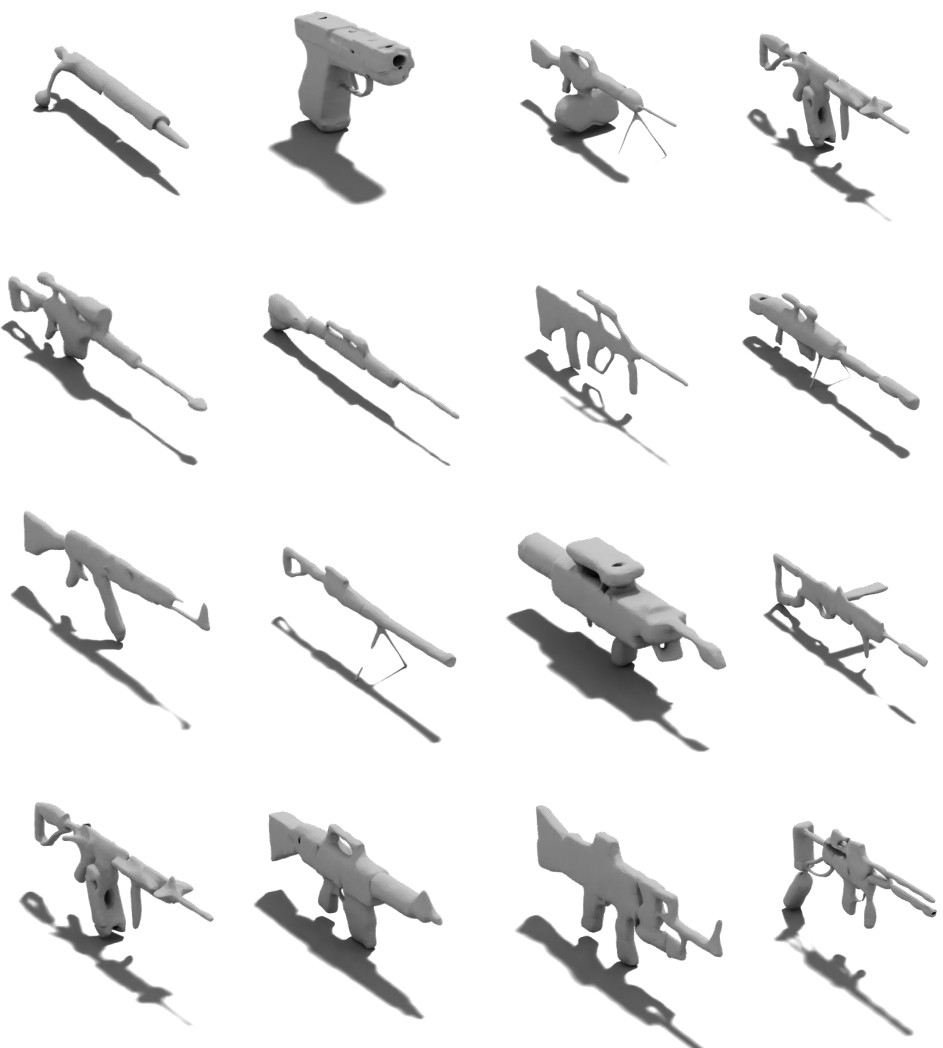

Figure 15: Generated 3D rifle meshes.

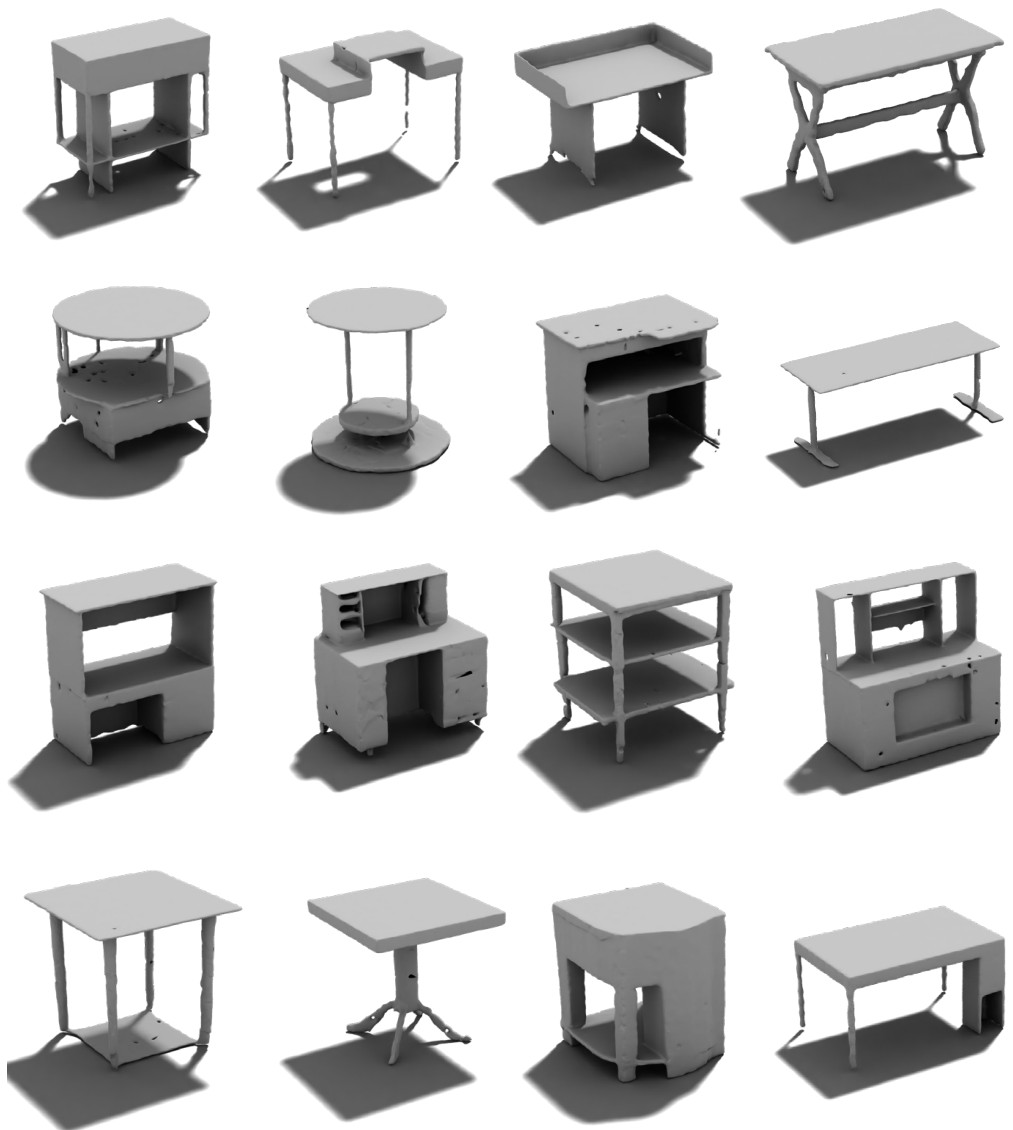

Figure 16: Generated 3D table meshes.

