# OpenReview forum: "MeshDiffusion: Score-based Generative 3D Mesh Modeling"
_ICLR.cc/2023/Conference — ICLR 2023 notable top 25%_

### Official Review · Reviewer_Mo7X · 2022-10-16

**Confidence:** 4
**Correctness:** 4
**Technical Novelty And Significance:** 3
**Empirical Novelty And Significance:** 3
**Recommendation:** 6

**Clarity, Quality, Novelty And Reproducibility:**

Clarity: The mathematical formulations in this paper needs more explanations. For the training objective equation in Sec. 4.2, it would be better if the symbols are explained right in this section, because it is hard for readers to find the meaning of the symbols in Sec. 3. Besides, the training objective equation is not numbered.

Quality: The result is better than compared methods. However, it is unclear whether it outperforms GET3D.

Novelty: The novelty is questionable since it is very similar to GET3D.

Reproducibility: It should be reproducible given the implementation details.



**Strength And Weaknesses:**

+ The tetrahedral grid representation is able to generate fine details of 3D shapes with arbitrary topology.

+ The mesh output can be easily used in graphics pipelines.

- The main problem is: There's a recently published paper  "GET3D: A Generative Model of High Quality 3D Textured Shapes Learned from Images" in NeurIPS 2022. This paper is released on Sep. 22, which is before the paper deadline of ICLR 2023 (Sep. 28). So it should be considered as "previous work". The GET3D paper uses a similar idea (tetrahedral representation) to perform a similar task. However, in the submission the authors didn't mention, review, or compare with GET3D.

- Multiple loss functions are used for training the network. However, the ablation study does not show the necessity of each loss function.

**Summary Of The Paper:**

This paper presents MeshDiffusion, a method for generating 3D meshes. The authors use the deformable tetrahedral grid as the 3D representation, and train a diffusion model on this parameterization. The authors demonstrated the effectiveness of the proposed model on multiple generative tasks, including unconditional and conditional generation.

**Summary Of The Review:**

The paper proposes a mesh generation model based on tetrahedral grid representation and diffusion model. The results show that the proposed method outperforms the baseline methods. However, the main issue is that one main state-of-the-art method "GET3D" was not mentioned or compared in this paper. Thus I would suggest the authors add this comparison and re-submit the paper.

---

> ### Author Response · Authors · 2022-11-19
> **Reply to Reviewer Mo7X from the authors**
>
> We appreciate the reviewer for raising the concerns.
>
> > The main problem is: There's a recently published paper "GET3D: A Generative Model of High Quality 3D Textured Shapes Learned from Images" in NeurIPS 2022. This paper is released on Sep. 22, which is before the paper deadline of ICLR 2023 (Sep. 28). So it should be considered as "previous work". The GET3D paper uses a similar idea (tetrahedral representation) to perform a similar task. However, in the submission the authors didn't mention, review, or compare with GET3D.
>
> We thank the reviewer for pointing out GET3D. We have included GET3D as well as few other concurrent papers in the related work section in our updated draft. We are currently training and evaluating our model with a higher resolution (as GET3D uses a higher resolution) as well as GET3D models with our dataset split for empirical comparison, and we will update the empirical results once we finish our model training.
>
> With all due respect, we would like to argue that a paper released only a week before ICLR deadline should be considered as “concurrent” instead of “previous” work. And therefore we strongly believe that the novelty and contribution of our paper should not be judged from sharing a similar goal to GET3D and the single question of “outperforming GET3D or not”.
>
> Furthermore, our paper uses different experiment settings from those of GET3D (we follow the same dataset split and settings with SDF-StyleGAN) and as a result the quantitative figures shown in GET3D are not directly comparable.
>
> More importantly, the code of GET3D was released only after the ICLR submission deadline and thus there was no possibility for us to make a proper comparison at that point.
>
> > The mathematical formulations in this paper needs more explanations. For the training objective equation in Sec. 4.2, it would be better if the symbols are explained right in this section, because it is hard for readers to find the meaning of the symbols in Sec. 3. Besides, the training objective equation is not numbered.
>
> We apologize for the confusion due to our writing. We have included some explanations for the symbols in Sec. 4.2 in the updated draft, and hopefully that will be clearer to readers.
>
> > Multiple loss functions are used for training the network. However, the ablation study does not show the necessity of each loss function.
>
> Only the dataset construction stage requires multiple loss functions, and most of the loss terms are covered by the original paper of DMTet [1]. Empirically the performance won’t differ too much with the choices of hyperparameters once the dataset attains a substantially low PSNR for rendered RGB image. The PSNRs of the reconstructed shapes averaged over 50 views are around 40, meaning that the visual quality of the dataset is good enough. For the effect of these losses on single shape reconstruction, please refer to the DMTet paper [1].
>
> Reference
>
> [1] Tianchang Shen, Jun Gao, Kangxue Yin, Ming-Yu Liu, and Sanja Fidler. Deep marching tetrahedra: a hybrid representation for high-resolution 3d shape synthesis. In NeurIPS, 2021.

---

> > ### Author Response · Authors · 2022-12-05
> > **Update on comparison with GET3D**
> >
> > We thank the reviewer for the patience. We have posted our new experiment results and show the quantitative comparison with GET3D in a separate comment. And please let us know if you have any remaining concerns.

---

> > > ### Comment · Reviewer_Mo7X · 2022-12-10
> > > **Response**
> > >
> > > Thanks for the effort. I'm changing my score given the updated result.

---

> > > > ### Author Response · Authors · 2022-12-10
> > > > **Thanks for reading our response!**
> > > >
> > > > We sincerely thank the reviewer for the comments and suggestions for improving our paper. We are very glad that the reviewer's concerns have been addressed. If there is any remaining concern or question, please feel free to let us know. We will be more than happy to address them.

---

### Official Review · Reviewer_NSt1 · 2022-10-24

**Confidence:** 5
**Correctness:** 3
**Technical Novelty And Significance:** 2
**Empirical Novelty And Significance:** 2
**Recommendation:** 6

**Clarity, Quality, Novelty And Reproducibility:**

The paper is well-structured and gives as many details as possible.
Architecture and hyperparameters are given. I think the paper is reproducible.

**Details Of Ethics Concerns:**

NO.


**Strength And Weaknesses:**

Strength:
- To my knowledge, it is the first diffusion model on 3D **mesh**.
- On unconditional generation evaluation, it truly beats one of the latest SOTA(SDF-StyleGAN).

Weakness:
- Not particularly novel. The paper is more like a combination of previous tetrahedral generation work and the diffusion model, especially when I see that the method has two training steps.

- Misses evaluation on SVR. Obviously, the first step can reconstruct shapes from 2D images. However, I could not find any evaluation on it neither in the paper nor in the appendix.

- Miss closest shape results when showing generated shapes. Can you show the nearest shapes in training set to validate the generation capability of your generative model?

- I saw claims like `extremely easy and stable to train.` and `MeshDiffusion is also very stable to train without bells and whistles.`. Is there any evidence to prove this?

- Holes seem still exist on the generated meshes. Are there any quantitative results that can support your point `REDUCING NOISE-SENSITIVITY OF MARCHING TETRAHEDRA`?

- Why FID metric of the car category is so high(even worse than IM-GAN)?

- Misses comparison with GET3D.

Misses references:
point cloud diffusion model:
LION: Latent Point Diffusion Models for 3D Shape Generation

recent tetrahedral mesh generation method:
GET3D: A Generative Model of High Quality 3D Textured Shapes Learned from Images

Typos:
`including unconditional generation, conditional generation and interpolation` ->
`including unconditional generation, conditional generation**,** and interpolation`

**Summary Of The Paper:**

The paper proposed a diffusion model-based mesh generator.
It adopts the representation from the previous `deep marching tetrahedral`.
The method works in two steps by first learning to reconstruct 3D meshes and then training a diffusion model on the grid points using a 3D U-Net.


**Summary Of The Review:**

The paper proposes the first diffusion model on 3D meshes.
However, due to its limited evaluation, I have doubts on its generation capacity and its claims on stable training and reducing noise.
In addition, quantitative comparisons in FID metrics with baseline methods can not show its superiority.
The paper also misses a comparison with recent GET3D.
So I am slightly leaning toward rejection.

---

> ### Author Response · Authors · 2022-11-18
> **Reply to Reviewer NSt1 from the authors (Part 1)**
>
> We thank the reviewer for spending time and raising the questions.
>
> > Not particularly novel. The paper is more like a combination of previous tetrahedral generation work and the diffusion model, especially when I see that the method has two training steps.
>
> While methodology-wise our model is more or less straightforward, we would like to argue that our model, along with GET3D and possibly LION, is one of the first to propose to directly unconditionally generate a parametrization of 3D meshes, and our model is the first diffusion model for such a purpose. Furthermore, we demonstrate that a naive diffusion model leads to much worse results (Sec. 4.3 and Table 3 in our paper). Our results imply that for diffusion models in mesh generation and possibly 3D in general, one needs to consider how the 3D shape is parametrized and design losses accordingly.
>
> > Misses evaluation on SVR. Obviously, the first step can reconstruct shapes from 2D images. However, I could not find any evaluation on it neither in the paper nor in the appendix.
>
> For the single RGBD view case, please refer to the conditional generation section in our paper. It is termed “conditional generation” instead of “reconstruction” because the shapes are not unique given one single view.
>
> For the RGB view case, it is theoretically possible to apply classifier-guidance method [1] for diffusion model inference so that an additional gradient term from the RGB single view supervision can be included. However, the rasterization process is highly nonlinear while the diffusion model inference process is very noisy. As a result, it can be very tricky to balance the gradient from the estimated score and that from the RGB supervision. Therefore, we do not include such a case in our paper.
>
> > Miss closest shape results when showing generated shapes. Can you show the nearest shapes in training set to validate the generation capability of your generative model?
>
> We show the pairs of generated samples and their corresponding nearest neighbor results in this anonymous google drive link (https://drive.google.com/file/d/1jFMtBpOHRUgPNWZ26--6Fnx33Nr96250/view?usp=share_link). These shapes, plus the interpolated examples shown in the paper, clearly show that our model generalizes beyond the training set.
>
> > I saw claims like extremely easy and stable to train. and MeshDiffusion is also very stable to train without bells and whistles. Is there any evidence to prove this?
>
> We claim our model is “easy to train” because
> - In theory, the score matching objective is a supervised one (the same as in denoising autoencoders). In contrast, the GAN objective is a min-max one, of which the optimization dynamics is much more complex.
>
> - Our diffusion model is trained with almost the same setting as the one used in [2], except that we use 3D convolution, and we do not observe any optimization issue.
>
> - We use the same set of hyperparameters for all categories, while in SDF-StyleGAN and GET3D, the hyperparameters differ (based on their github official code).
>
> > Holes seem still exist on the generated meshes. Are there any quantitative results that can support your point REDUCING NOISE-SENSITIVITY OF MARCHING TETRAHEDRA?
>
> Please see “Ours, No Discretization” in Table 3 for the quantitative results in our original draft. We aim to reduce the error due to nonlinearity in marching tetrahedra by setting SDFs to +1/-1, and thus we make the distinction between “non-discretized” (the naive objective) and the “discretized” (proposed) version. We apologize for the unclear connection between Sec. 4.3 and the corresponding empirical results. We have changed “discretized SDF” to “normalized SDF” in our updated draft as we found it more intuitive.
>
> > Why FID metric of the car category is so high(even worse than IM-GAN)?
>
> First, the FID scores are computed by an average of FID scores on rendered images, while these images are rendered with a uniform diffuse color with simple point light sources. The distribution of these images are wildly different from that of natural images (which ImageNet-pretrained models used for FID are trained on). Therefore the “FID scores” in our setting is not a stronger indicator for 3D geometry quality. We have to take all the quantitative metrics into consideration when evaluating the generation quality of MeshDiffusion’s 3D shapes.
>
> Second, the local geometry is generally much more complicated (seats, visible inner structures, etc.) and therefore a higher resolution is required to fully capture these details. Our original model is trained with a resolution of 64 and as a result the dataset is much noisier. We qualitatively found that a higher resolution of 128 can largely solve this issue. We will update the detailed quantitative number once we finish the training.

---

> > ### Author Response · Authors · 2022-11-18
> > **Reply to Reviewer NSt1 from the authors (Part 2)**
> >
> > > “Misses comparison with GET3D.” and “Misses references: point cloud diffusion model: LION: Latent Point Diffusion Models for 3D Shape Generation”
> >
> > We thank the reviewer for pointing out these important reference papers. We have mentioned these papers in the related work section in our updated draft. For empirical comparison, we are currently training and evaluating our model with a higher resolution (as GET3D uses a higher resolution) as well as GET3D models with our dataset split. We will release the results later within the rebuttal period and include them in our draft.
> >
> > While we will include these new results, we would like to underscore that our paper is concurrent with GET3D (paper released one week before ICLR submission deadline and code available only after it), and LION is released on Arxiv in October. Therefore, we argue that our paper should not be judged solely based on the similarities and comparison with GET3D and LION.
> >
> > > Typos: including unconditional generation, conditional generation and interpolation -> including unconditional generation, conditional generation**,** and interpolation
> >
> > Thanks for your suggestion. We have fixed this point as well as some other typos we found so far in our updated draft.
> >
> > Reference
> >
> > [1] Ho, Jonathan, and Tim Salimans. "Classifier-free diffusion guidance." arXiv preprint arXiv:2207.12598 (2022).
> >
> > [2] Yang Song, Jascha Sohl-Dickstein, Diederik P Kingma, Abhishek Kumar, Stefano Ermon, and Ben Poole. Score-based generative modeling through stochastic differential equations. In ICLR, 2021.

---

> > > ### Author Response · Authors · 2022-12-05
> > > **Update on comparison with GET3D**
> > >
> > > We thank the reviewer for the patience. We have posted our new experiment results and show the quantitative comparison with GET3D in a separate comment. And please let us know if you have any remaining concerns.

---

> > > > ### Comment · Reviewer_NSt1 · 2022-12-07
> > > > **Reply to the authors**
> > > >
> > > > Happy to see the comparison with GET3D and the visualization of the inner structures in the car category, which alleviate my concerns. For now, I change my rating to weak accept. Another request: could you show the closest shapes in the training set for your unconditional generation results in `https://drive.google.com/file/d/1ib4LxxvCiF-tXwDQJ8y0NrVb9Ry33UAC/view`. I believe it can make the paper more convincing.

---

> > > > > ### Author Response · Authors · 2022-12-09
> > > > > **Nearest-neighbor visualization**
> > > > >
> > > > > We thank the reviewer for acknowledging our contribution and suggesting the nearest neighbor comparison for our new results. Please find a sample set of pairs of generated meshes from our resolution-128 models and their corresponding nearest neighbors in https://drive.google.com/file/d/1SgYner50E85J_sV_BODia7NIyslKXl4j/view?usp=share_link.

---

### Official Review · Reviewer_jFa3 · 2022-10-26

**Confidence:** 3
**Correctness:** 4
**Technical Novelty And Significance:** 2
**Empirical Novelty And Significance:** 2
**Recommendation:** 6

**Clarity, Quality, Novelty And Reproducibility:**

The paper generally reads okay, kind of clear if you are familiar with the score-based method. Figure 2 is very similar to the original deep marching tetrahedra paper (not copied but does look similar). For general audience especially those who are not familiar with score-based method the paper might be a little hard to read, at least in the beginning. Novelty wise the paper seems is a combination of two frameworks the deep marching tetrahedra and the score based method, but applies to a new field of 3D mesh generation, in this perspective it can be considered novel. I am not sure about how easier to reproduce the code yet.

**Strength And Weaknesses:**

The paper is for 3D mesh generation using the score based diffusion model. Score based diffusion model for generative modeling such as image synthesis has shown significant improvement over existing methods such as GAN, VAE, etc. This paper tries to extend the score based diffusion method into 3D mesh generation domain, which is a very important area to explore.

The main framework is based on Deep Marching Tetrahedra, plus the score based diffusion model. These are the main strength of the proposed method which essentially combines these two and applies to mesh generation.

The results seem good in comparison with existing methods.

The work on the other hand, seems still at a very earlier stage, and the paper seems generated in a little rush, e.g. there are two citations of 38,38] in page 3, second line, second paragraph. There are probably still some issues that need to solve before it can be fully utilized, as some of the limitations mentioned by the authors.

Nonetheless, this paper does point to an interesting direction in 3D mesh generation which the 3D community might not be fully aware of before (including myself), in this point the paper has merit and is to be accoladed.


**Summary Of The Paper:**

This paper proposes a diffusion model MeshDiffusion for 3D meshes generation.  The meshes are embedded in a deformable
tetrahedral grid, and then employ a score based method to train a diffusion model on this direct parameterization. Besides using in the 3D space, the network can be used when only 2.5D information is available as well. The model is tested  on multiple generative tasks.

**Summary Of The Review:**

Overall it is an interesting application of the score based method in the 3D mesh generation domain which is very important. The paper does show some good results over the existing method. The paper would benefit from more time spent on the presentation, illustration, proof reading etc as it contains some typos and errors.

---

> ### Author Response · Authors · 2022-11-18
> **Reply to Reviewer jFa3 from the authors**
>
> We thank the reviewer for acknowledging our contribution and apologize for our unclear writing and typos. We have updated our draft and fixed the typos we found so far. For reproducibility, we will release our code and trained models.

---

> > ### Author Response · Authors · 2022-12-05
> > **Update on new experiment results**
> >
> > We thank the reviewer again for the time and efforts. To address other reviewers' concerns over the performance comparison against GET3D, a concurrent mesh generation method, we have posted our new experiment results in a separate comment. And please let us know if you have any remaining concerns.

---

### Author Response · Authors · 2022-12-05
**Quantitative Comparison with GET3D**

We thank the reviewers again for their time, efforts and patience. We have completed the experiments and show the performance comparison between our MeshDiffusion and GET3D on Chair and Car categories in the table below:

---
### Chair

Method | MMD-CD | MMD-EMD | MMD-LFD | COV-CD | COV-EMD | COV-LFD
-------------------|------------------|------------------|------------------|------------------|------------------|------------------
MeshDiffusion |  **0.003329** | 0.08702 | **3534** | **73.60** | **73.97** | 61.73
GET3D | 0.003435 | **0.08595** | 3665 | 71.98 | 71.39 | **68.81**

---
### Car

Method | MMD-CD | MMD-EMD | MMD-LFD | COV-CD | COV-EMD | COV-LFD
-------------------|------------------|------------------|------------------|------------------|------------------|------------------
MeshDiffusion |  **0.001268** | **0.05465** | **2444** | **51.09** | **54.17** | **57.53**
GET3D | 0.001703 | 0.06292 | 2600 | 22.47 | 31.56 | 47.07

More specifically, we train our MeshDiffusion on our ShapeNet split with DMTet grid of resolution 128. We note that this new setting matches the adaptive resolution in GET3D, and it is therefore a fair comparison. We generate a set of 5x test set size for the computation of MMD and Coverage, the same as in the GET3D paper.

The point cloud metrics (CD and EMD) of GET3D on the “Car” category is significantly worse because the ground truth meshes contain inner structures (seats, etc.) which are invisible from the outside. Because GET3D does not provide any 3D supervision, it is not able to attain a good score on naive point cloud metrics. Therefore, we run the GET3D official code in https://github.com/nv-tlabs/GET3D/tree/master/evaluation_scripts to sample point clouds only on the outer surfaces, and recompute the point cloud metrics on these outer-surface-only point clouds. The figures are shown below.

---
### Car (outer surface only; point cloud metrics)

Method | MMD-CD | MMD-EMD | COV-CD | COV-EMD
-------------------|------------------|------------------|------------------|------------------
MeshDiffusion |  **0.000680** | **0.03668** | **64.13** | **68.80**
GET3D | 0.000985 | 0.03997 | 42.93 | 54.20


For qualitative improvement by using a higher resolution, we show the samples in the following anonymous link: https://drive.google.com/file/d/1ib4LxxvCiF-tXwDQJ8y0NrVb9Ry33UAC/view?usp=share_link. Some of the examples impossible in daily life show the generalization capability of MeshDiffusion, and some examples of generated car meshes show that MeshDiffusion, trained with 3D information, is able to generate concave and complex geometries. We further give an example of the invisible inner structures MeshDiffusion produces in https://drive.google.com/file/d/1NyU7VwtWIrgcDaWPxTWhcbrrYX0ptxRA/view?usp=share_link. In comparison, GET3D does not produce good inner structures due to lack of 3D supervision (example in https://drive.google.com/file/d/1yrkB4fdm-fQREm_ruuHEPHEjv8DA4OY-/view?usp=share_link).

---

### Decision · Program_Chairs · 2023-01-20

**Decision:**

Accept: notable-top-25%

**Justification For Why Not Higher Score:**

Although the work is essentially concurrent with GET3D, particularly in terms of comparison, it remains true that the core ideas are less surprising now that GET3D is known.

**Justification For Why Not Lower Score:**

Strong agreement between reviewers, even if all "weakly accepting" is a good indicator that the work is of interest to more than just a small subset of the audience.

**Metareview: Summary, Strengths And Weaknesses:**


The paper extends score-based diffusion models to 3D mesh generation, via direct parameterization on a tetrahedral grid.  All reviewers note the novelty of the idea, albeit co-invented with GET3D.

The authors were assiduous in producing a comparison to GET3D for the rebuttal, as well as answering other reviewer questions, and reviewers engaged with the authors' rebuttal, all converging on accepting the paper.



**Note From Pc:**

if the above contains the word "oral" or "spotlight" please see: "oral" presentation means -> notable-top-5% and "spotlight" means -> notable-top-25%. As stated in our emails, we are disassociating presentation type from AC recommendations